# High Concentration of Heavy Metal and Metalloid Levels in Edible *Campomanesia adamantium* Pulp from Anthropic Areas

**DOI:** 10.3390/ijerph18115503

**Published:** 2021-05-21

**Authors:** David Johane Machate, Elaine S. de Pádua Melo, Daniela G. Arakaki, Rita de Cássia Avellaneda Guimarães, Priscila Aiko Hiane, Danielle Bogo, Arnildo Pott, Valter Aragão do Nascimento

**Affiliations:** 1Graduate Program in Sciences of Materials, Federal University of Mato Grosso do Sul, Campo Grande 79079-900, Brazil; 2Group of Spectroscopy and Bioinformatics Applied Biodiversity and Health (GEBABS), Graduate Program in Science of Materials, Federal University of Mato Grosso do Sul, Mato Grosso do Sul 79070-900, Brazil; elainespmelo@hotmail.com (E.S.d.P.M.); daniarakaki@gmail.com (D.G.A.); 3Graduate Program in Health and Development in the Central-West Region of Brazil, Federal University of Mato Grosso do Sul, Campo Grande 79079-900, Brazil; rita.guimaraes@ufms.br (R.d.C.A.G.); priscila.hiane@ufms.br (P.A.H.); daniellebogo@hotmail.com (D.B.); 4Graduate Program in Biotechnology and Biodiversity in the Central-West Region of Brazil, Federal University of Mato Grosso do Sul, Campo Grande 79079-900, Brazil; arnildo.pott@gmail.com

**Keywords:** Cerrado, myrtaceae, edible fruit, farm-margin, roadside, macro- and microelements, health risk

## Abstract

This study aimed to quantify the extent of heavy metal, non-metal and metalloid levels in the *Campomanesia adamantium* pulp obtained from an area crossed by road experiencing high large vehicle traffic and intensive agriculture modern farming, to monitor the health risks associated with pulp consumption by humans. For this purpose, in three spots located within this area, ripe fruits were collected on the roadside, bush and farm-margin. Pulp samples were digested by microwave-assisted equipment, and chemical elements were quantified by ICP OES. The concentrations of K, Pb, Se, Fe, Mo, Zn, Co, Ni and Mn in the pulp collected in roadside/bush points showed statistical differences (*p* < 0.05). The heavy metals and metalloid concentrations that exceeded FAO/WHO standards were ordered Pb > As > Mo > Co > Ni > Mn > Cr. Therefore, among these metalloid and heavy metals, As, Pb and Cr were found to be higher in farm-margin > roadside > bush (1.5 × 10^−3^, 1.1 × 10^−3^ and 6.2 × 10^−4^), respectively. Therefore, As is the most important metalloid with higher levels in farm-margin, roadside and bush (1.5 × 10^−3^, 1.0 × 10^−3^ and 6.0 × 10^−4^ > 10^−6^–10^−4^ and 3.33, 2.30 and 1.34 > 1), respectively, to total cancer risk and hazard quotient, if 10 g daily of pulp are consumed.

## 1. Introduction

The relationship between anthropic activities and native fruits is extremely important for food security, including the role of metalloids and heavy metals in the contamination of land, water, and edible plants, which has been regarded as an environmental and public health hazard [1]. Due to severe anthropogenic activities, as high large vehicle traffic and intensive modern agriculture, the environment becomes prone to high toxicity and the bioaccumulation of heavy metals in plants used for food or medicines [2,3,4]. Among several species of plants, *Campomanesia adamantium* (Cambess.) O. Berg (Myrtaceae), popularly known as “Guavira or Guabiroba”, stands out for its wide occurrence in the Cerrado and other biomes, such as those of the Atlantic Forest and Pampa in Brazil, which have intensive and intense anthropogenic activities [5]. In addition, roots, leaves and fruits of this species are popularly used as antirheumatic, antidiarrheal, hypocholesterolemic, anti-inflammation, urethritis and cystitis remedies, among other functions [6,7,8]. Moreover, several studies have reported the potential activities of *C. adamantium* fruits as antibacterial and antifungal [9,10], anti-hyperalgesic, antidepressive [11], antimicrobial [12], antiproliferative against cancer cells [13,14], hepatoprotective [15], as an inhibitor of leukocyte mobility, neurogenic pain and oedema [7]. The genus *Campomanesia* includes 37 species, 26 of which are endemic in Brazil [5]. The *C. adamantium* fruits, characterized by a citrus aroma and sweet flavor, are consumed fresh or used to produce homemade liqueurs, juices, ice creams, jellies, backer products, and others [16]. Additionally, they are natural sources of a considerable amount of ascorbic acid, fibers, vegetable oil, polyphenols, and monoterpene substances [7,14,17].

To date, there are only studies that have quantified minerals in the peel, pulp and seed of *C. adamantium* collected near urban areas [18,19]. However, no studies have been carried out to assess the chemical elements in fruits collected close to roads with high vehicle traffic in agricultural regions. Fertilizers, pesticides, and vehicle fumes contain heavy metals and metalloids, such as potassium (K), arsenic (As), iron (Fe), lead (Pb), chromium (Cr), manganese (Mn), molybdenum (Mo), nickel (Ni), and other elements, which in high amounts contaminate the environment, edible plants, and consequently, humans [2,4,20,21].

In this context, using inductively coupled plasma optical emission spectroscopy (ICP OES), this study aimed to quantify potassium (K), lead (Pb), phosphorus (P), arsenic (As), selenium (Se), iron (Fe), molybdenum (Mo), zinc (Zn), cobalt (Co), nickel (Ni), manganese (Mn), and chromium (Cr) in the *C. adamantium* fruit pulp collected in these three spots from the roadside (500 m) to bush (1000 m) and farm-margin (3000 m), marked by intense anthropogenic activities. The concentrations of these chemical elements were compared to the recommended tolerable maximum intake levels established by Dietary Reference Intakes (RDI) for children aged 4–8 years, adults and pregnancy (31–50 years), and the Food Agriculture Organization of the United Nations (FAO) and World Health Organization (WHO) parameters for human intake. According to the Food and Drug Administration (FDA) parameters, the contents of the chemical elements in this pulp were qualified as a good source, excellent source or no good source. To the best of our knowledge, this is the first report on high concentrations of metalloids such as As and heavy metals like Pb and Cr in the pulp of a wild edible plant collected near high vehicle traffic and farming with the intensive use of fertilizers and pesticides. According to the carcinogenic risk calculated to health risk assessment, we propose that individuals consume 1 g/day instead of the 400 g/day—as recommended by WHO for edible fruits and vegetables—due to the high concentrations of As associated with several types of cancer and other diseases.

## 2. Materials and Methods

### 2.1. Fruit Collection and Sample Preparation

Ripe fruits were collected in twenty-one different points, separated from each other by 20 m. The fruits were mixed according to the collected distance from the roadside (500 m) to the bush (1000 m) and farm-margin (3000 m) in Campo Grande, Mato Grosso do Sul state, Brazil, 20°46′34.208″ S, 54°10′28.567″ W (Figure 1), in November 2019. Manually, the pulp was separated from the peel and seed, immediately dried in an air circulation oven at 40 °C for 48 h. The dried pulp was milled using mortar and pestle and sieved to obtain the refined powder, placed into an amber and hermetic glass bottle and frozen at −20 °C for further analyses.

The *C. adamantium* was registered in System of Genetic Resource Management and Associated Traditional Knowledge (SisGen) of the Ministry of the Environment (registration number A7716EC).

### 2.2. Microwave-Assisted Digestion Procedure

The pulp samples were weighed according to Lima et al. [19] and prepared as described: 0.5 g sample plus 5 mL HNO_3_ (65% Merck, Darmstadt, Germany) and 3 mL H_2_O_2_ (35% Merck, Darmstadt, Germany) were individually placed into PTFE bottles of the DAP 60 type (Berghof). The mixture was allowed to remain in the open air for 10 min predigestion and then digested using a microwave digestion system (Speedwave four^®^, Berghof, Germany). After the microwave system’s digestion procedure, the samples were transferred from the vessels to 50 mL Falcon vessels and which were then filled to 30 mL with ultrapure water (conductivity 18.2 MΩcm (Millipore), Biocel, Germany). The samples were digested in the microwave system according to the schedule shown in Table 1. All the digestion analysis steps were performed in triplicate.

### 2.3. ICP OES Elemental Analysis

Chemical elements were quantified using the ICP OES (Thermo Fischer Scientific, Bremen, Germany, iCAP 6300 Duo) technique. The selected emission lines (wavelength in nm) for determining elements in pulp and operating conditions of ICP OES are summarized in Table 2.

### 2.4. Calibration Curves

For the ICP OES, standard solutions for analytical calibration were prepared by diluting a standard multiple-element stock solution containing 1000 mg/L of the Al, As, Ca, Cd, Co, Cr, Cu, Fe, Mg, Mn, Mo, Na, Ni, P, S, V, Se, and Zn from SpecSol (SpecSol, Quimlab, Brazil). For each element detected, the limit of detection (LOD) of 0.0002–0.003 (mg/L), the limit of quantification (LOQ) of 0.006–0.01 (mg/L) and the correlation coefficient (R^2^) of 0.9995–0.9998 were determined. One blank and seven calibration curves were generated using the following concentrations: 0.01, 0.02, 0.05, 0.2, 1.0, 2.0 and 5.0 mg/L of the element standard. All experiments were carried out in triplicate. The detection limit (LOD) was calculated as three times the standard deviation of the blank signal (B) expressed in concentration divided by the slope of the analytical curve (AC): LOD = 3*B/AC, and the limit of quantification (LOQ) was obtained as ten times the standard deviation of the blank divided by the slope of the analytical curve: LOQ = 10*B/AC [22].

An addition/recovery test for the elements under study was performed in a pulp sample by spiking 0.5 mg/L of each analyte. The method had a recovery interval of 80%–110% for the spike 0.5 mg/L, which was found to be between 70% and 120% to the established limit proposed by the Union of Pure and Applied Chemistry (IUPAC) and Association of Official Analytical Chemists (AOAC) [23,24].

### 2.5. Human Health Risk Assessment

The results of the concentrations of the chemical elements were compared with recommended intake standards of the RDA/AI, UL, FAO/WHO, USEPA and hazard quotient. The human risk for a non-carcinogenic was calculated following the equation adopted by Liang et al. [25]. Cancer risk was the probability of an individual developing any cancer over a lifetime, during the daily doses exposure to 70 years; the chronic daily intake dose (CDI) of carcinogenic elements (mg/kg/day); and slope factor (SF) was the carcinogenicity (mg/kg/day). The SFs of As, Cr and Pb are 1.5, 0.5 and 0.0085 mg/kg/day, respectively, following Equation (1):(1)Cancer Risk=CDI×SF

The cancer risk is a sum of individual variety carcinogenic elements risk in different exposure pathways, which is the total cancer risk (R). In agreement with USEPA [26], the value of acceptable or tolerable cancer risk ranges from 10^−6^ to 10^−4^, while > 10^−4^ is considered unacceptable.

The human health risk of heavy metal intake was evaluated based on the chronic daily intake dose (CDI, mg/kg/day) for a chemical contaminant in the pulp over the exposure period and the pulp intake quantity. CDI was calculated using the following Equation (2):(2)$${\mathrm{CDI}}_{\mathrm{pulp}}=\frac{C_{\mathrm{pulp}}\times {\;\mathrm{IR}}_{\mathrm{pulp}}\times \mathrm{EF}\times \mathrm{ED}}{\mathrm{BW}\times \mathrm{AT}}$$
where CDI_pulp_—chronic daily pulp intake dose; C_pulp_—concentration of chemical element content in the pulp; IR_pulp_—ingestion rate (mg/day); EF—exposure frequency (90 days available/year); ED—exposure duration (life expectancy = 70 years); BW—body weight; and AT—average time (ED × 365 days). The adult’s body weight, approximately 70 kg, and the average daily pulp consumption was 10 g/day. The risk to human health by the intake of heavy metal-contaminated pulp was measured using a hazard quotient (HQ), which is a ratio of CDI and chronic oral reference dose (RfD), determined by the following Equation (3):(3)$$\mathrm{HQ}=\frac{\mathrm{CDI}}{\mathrm{RfD}}$$

The RfD values for the risk calculation were established by the Joint Food and Agriculture Organization/World Health Organization Expert Committee on Food Additives [27] and the United States Environmental Protection Agency [28]. The RfD values for the elements were established: K = not available; Pb = 0.004 mg/kg bw/day; P = not available; As = 0.0003 mg/kg bw/day; Se = not available; Fe = 0.7 mg/kg bw/day; Mo = 0.005 mg/kg bw/day; Zn = 0.3 mg/kg bw/day; Co = 0.03 mg/kg bw/day; Ni = 0.02 mg/kg bw/day; Mn = 0.14 mg/kg bw/day; and Cr = 0.003 mg/kg bw/day [28]. As shown in Equation (3), a toxic risk is considered to occur if HQ *>* 1, whereas HQ < 1 represents a negligible hazard (adverse non-carcinogenic effects).

### 2.6. Statistical Analysis

The data were analyzed by one-way ANOVA using the GraphPad Prism software version 8.0 for Mac (GraphPad Software, San Diego, CA, USA). The significance of the differences between the means for the individual element level was considered at *p* < 0.05.

## 3. Results and Discussion

In this section, the article was composed of two Sections: Section 3.1 present data on the concentration of the chemical elements obtained in pulp collected in roadside, bush and farm-margin, and the comparison of these concentrations with other published studies. In Section 3.2, data of the type of chemical elements quantified for each site was used to calculate EDI and HQ values.

### 3.1. The Chemical Element Concentration in Pulp Collected in Three Different Sites

Twelve chemical elements were found in *C. adamantium* pulp collected in three different sites from the road: roadside (500 m); bush (1000 m); and farm-margin (3000 m) (Table 3).

The concentration of chemical elements quantified in *C. adamantium* pulp samples is depicted in decreased order in Table 3. The average concentration of chemical elements in pulp collected on roadside followed in decreased order K > Pb > P > As > Fe > Se > Mo > Zn > Co > Ni > Mn > Cr; bush: K > Pb > P > As > Se > Fe > Mo > Zn > Ni > Mn > Co > Cr, and farm-margin: K > Pb > P > As > Se > Fe > Mo > Zn > Co > Ni > Mn > Cr. The concentrations of Pb, As and Cr in the present study are higher compared with the average reported for fruits and pulp collected in areas with a lower exposure to contaminants produced by anthropogenic activities [18,19,34], that exceed the FAO/WHO permissible limit recommended for edible berries and small fruits [29,30,31,32,33]. On the other hand, high concentrations of Mo, Co, Ni and Mn were reported in *C. adamantium* fruits compared with the present study [34], which could be correlated with the occurrence of these chemical elements in natural environments [35,36].

In general, the average of all chemical elements quantified in *C. adamantium* pulp followed a decreasing order K > Pb > P > As > Se > Fe > Mo > Zn > Co > Ni > Mn > Cr. The one-way ANOVA test values considering the concentrations of each element were in collected in the three sites; then, we compared the pairs roadside/bush, roadside/farm-margin and bush/farm-margin. The concentrations of K, Pb, Se, Fe, Mo, Zn, Co, Ni and Mn in the pulp collected in roadside/bush points showed statistical differences (*p* < 0.05). However, significant differences (*p* > 0.05) were not observed when comparing the concentration of each chemical element found in *C. adamantium* pulp collected in roadside/bush/farm-margin.

Thus, it was observed that the concentration behavior of chemical elements decreased from the roadside (500 m) to bush (1000 m) and increased to farm-margin (3000 m). However, the concentrations of Pb and Se increased from the roadside to the bush and more toward the farm-margin, as illustrated in Figure 2.

Table 3 list the levels of chemical elements quantified (mg/100 g ± SD) in the *C. adamantium* pulp compared with the limit specification of RDAs/AI and UL values for males and females (31–50 y), pregnant women (31–40 y) and children (4–8 y) [37], and FAO/WHO and WHO [29,30,31,32,33] permissible levels for fruits and food.

The percentages of chemical elements in the pulp were calculated from the mean values (Table 3) based on RDA, AI, UL, and FAO/WHO and WHO limits [29,30,31,32,33], while the studied chemical elements were qualified based on the FDA parameters (10%–19% for “good source” of nutrition, and ≤20% for “excellent source”) [38].

Potassium (K) concentrations in roadside (33.02 ± 0.01 mg/100 g), bush (31.02 ± 0.01 mg/100 g) and farm-margin (58.01 ± 1.34 mg/100 g) pulp correspond to proportions ≤ 1% for males, females, pregnant women, and children to 4700 mg/day by RDA parameters. The UL of K has no established values for males, females, pregnant women and children. The K content in this pulp was the lowest (3510 mg/day) FAO/WHO limit [29]. According to FDA parameters, this pulp is not a good source of K [38]. The K concentrations in this pulp are lower than 130–253 mg/100 g, as reported in previous studies for *C. adamantium* fruits and pulp [18,19,34], which can be explained by the higher levels of metalloid and heavy metals observed in this pulp which can reduce K content, such as Cr, which results from intense anthropogenic activity [39]. However, K concentrations in this pulp are near blueberry and alfalfa (39 mg/100 g) [38]. The health benefit of K in the body is associated with blood pressure regulation, stroke risk reduction, preventing renal system dysfunction, decreasing urinary calcium excretion, reducing kidney stone formation and osteoporosis disease [40], regulating blood lipid concentrations [36] and maintaining bone and cardiovascular health [41,42,43].

Lead (Pb) concentrations in roadside (5.36 ± 0.02 mg/100 g), bush (7.02 ± 0.01 mg/100 g) and farm-margin (7.88 ± 1.05 mg/100 g) pulp correspond to 26,800%, 35,100% and 39,480% by 0.02 mg/day FAO/WHO parameters [30]. The RDA and UL parameters for Pb have no established values for adults and children. Based on the FDA parameters, this pulp is an excellent source of Pb [38]. In this pulp, Pb concentrations are lower than 0.06 mg/100 g, as reported in previous studies for fruits of *C. adamantium* [18]. On the other hand, Pb contents in this pulp are near those of other edible fruit such as apple (2.35 mg/100 g), mango (6.72 mg/100 g) [44] and tomato (5.41–11.73 mg/100 g) [45]. The risk of consuming food with a high amount of Pb is correlated with intelligence reduction, bone joint weakness, accelerated bone maturation, increased blood pressure, spontaneous abortion, renal dysfunction, allergic diseases [46], respiratory and cardiovascular diseases [47].

Phosphorus (P) concentrations in roadside (3.24 ± 0.02 mg/100 g), bush (3.04 ± 0.02 mg/100 g) and farm-margin (5.24 ± 0.8 mg/100 g) pulp correspond to proportions ≤ 1% for males, females and pregnant women (700 mg/day) and children (500 mg/day) by RDA parameters. The P contents correspond to values ≤ 0.2% for males/females (4000 mg/day), pregnant women (3500 mg/day) and children (3000 mg/day) by UL limits. The P concentrations of the roadside, bush and farm-margin pulp correspond to proportions < 1% to 700 mg/day by FAO/WHO limits [29,31]. According to FDA parameters, this pulp is not a good source of P [31]. Indeed, P concentrations in this pulp are lower than 17–196 mg/100 g reported in previous studies on fruits and pulp of *C. adamantium* [18,19,34]. However, P concentrations in this pulp are near of blackberry and watermelon (5–11 mg/100 g) [38]. The health benefit of P consumption is related to bone mineralization, cell energy generation, cardiovascular regulation and neuromuscular function [48], and the modulation of short-chain fatty acid gut bacteria producers [49].

Arsenic (As) concentrations in roadside (1.96 ± 0.04 mg/100 g), bush (1.14 ± 0.03 mg/100 g) and farm-margin (2.84 ± 0.52 mg/100 g) pulp correspond to 19,600%, 11,400% and 28,400% by 0.01 mg/day FAO/WHO limits [30]. The RDA and UL parameters for As have no established values for adults and children. By FDA parameters, this pulp is an excellent source of As [38]. The As contents in this pulp are higher than 0.07 mg/100 g, as reported in previous studies on *C. adamantium* fruits [18] and are near those of edible vegetables such as lettuce (2.73 mg/100 g) [8], *Colocasia antiquorum* (0.6–12.5 mg/100 g), gourd leaf (0.8–15.8 mg/100 g) [50], fish, seafood and seafood products (0.16–0.56 mg/100 g) [51]. The risk of the consumption of food with a high amount of As is associated with cancers (skin, lung and bladder) [50], respiratory disease, gastrointestinal disorder, liver malfunction, neuro–cardiovascular dysfunction, anemia disorder, leucopenia and thrombocytopenia effects, diabetes [52], cytotoxicity and genotoxicity effects [53].

Selenium (Se) concentrations in roadside (0.20 ± 0.01 mg/100 g), bush (0.22 ± 0.02 mg/100 g) and farm-margin (0.40 ± 0.1 mg/100 g) pulp correspond to values <1% for males and females (55 mg/day), pregnant women (400 mg/day) and children (30 mg/day) by RDA parameters. The Se contents in the roadside, bush and farm-margin pulp correspond to proportions of <1% for males and females (400 mg/day), pregnant women (60 mg/day) and children (150 mg/day) by UL limits. The Se concentrations in roadside, bush and farm-margin pulps correspond, respectively, to 333.33%, 366.67% and 500% by 0.06 mg/day FAO/WHO limits [31]. According to FDA parameters, this pulp is an excellent source of Se [38]. The Se concentrations in this pulp are lower than the amount of 0.88 mg/100 g reported in previous studies on *C. adamantium* fruits [18], and higher than that reported in grapes, apricot, kiwi, litchi, macadamia and pistachio (0.0001–0.007 mg/100 g) and near that of the cashew nut (0.02 mg/100 g) [38]. Other studies have recommended 0.018 mg/day of Se quantity intake for children (4–6 y), 0.023 mg/day for adolescent males 10–18 y and 0.021 mg/day for adult females (19–65 y), 0.027 mg/day for males and 0.0204 mg/day [54]. The benefit of the consumption of Se is correlated with preventing and decreasing diabetes mellitus, cancers [55], improving male fertility [56,57], human neuropathies [58] and hepatic steatosis [59].

Iron (Fe) concentrations in roadside (0.23 ± 0.02 mg/100 g), bush (0.12 ± 0.01 mg/100 g) and farm-margin (0.40 ± 0.10 mg/100 g) pulp correspond to values <4% by RDA parameters for males (8 mg/day), females (18 mg/day), pregnant women (27 mg/day) and children (10 mg/day). The Fe contents in the roadside, bush and farm-margin pulp correspond to <1% by UL parameters for males, females and pregnant women (45 mg/day) and children (40 mg/day). In concordance with FDA parameters, this pulp is not a good source of Fe [38]. The Fe concentrations in this pulp are lower than the amount of 1–2.6 mg/100 g reported in previous studies on fruits and pulp of *C. adamantium* [18,19,34]. However, the Fe content of this pulp is between that of apple, guava and pineapple (0.12–0.29 mg/100 g) [38]. The health benefits of food consumption with Fe are improving maximal oxygen respiration and exercise performance, hemoglobin synthesis, electron transport, anemia prevention, deoxyribonucleic acid synthesis, gut microbiota modulation, neurodevelopment, immunity, pregnancy development [60,61,62].

Molybdenum (Mo) concentrations in roadside (0.10 ± 0.02 mg/100 g), bush (0.09 ± 0.02 mg/100 g) and margin-farm (0.19 ± 0.01 mg/100 g) pulp correspond to proportions ≤1% by RDA parameters for males and females (150 mg/day), pregnant women (50 mg/day) and children (22 mg/day). The Mo contents in the roadside, bush and farm-margin pulp correspond to values ≤0.2% by UL parameters for males and females (1100 mg/day), pregnant women (2000 mg/day) and children (600 mg/day). The Mo concentrations in roadside, bush and farm-margin pulp correspond, respectively, to 222.22%, 177.78% and 422.22% by 0.045 mg/day FAO/WHO parameters [31]. In agreement with FDA parameters, this pulp is an excellent source of Mo [38]. However, the Mo concentrations in this pulp are lower than the amount of 0.4–30 mg/100 g reported in previous studies on fruits of *C. adamantium* [19,34]. The Mo food consumption is recommended for infants (0.015–0.04 mg/day) and all individuals ≥10 years old (0.025–0.15 mg/day) [63]. The health benefit of Mo is correlated with toxicity prevention by several metabolites, reduction in aerosol organs irritability, night blindness, neurological damage, aches and pain [64,65,66]. The Mo concentrations of this pulp are between those of pea seeds and tomato (0.10–0.20 mg/100 g) [67].

Zinc (Zn) concentrations in roadside (0.08 ± 0.01 mg/100 g), bush (0.07 ± 0.01 mg/100 g) and farm-margin (0.13 ± 0.02 mg/100 g) pulp correspond to values <2% by RDA limits for males and pregnant women (11 mg/day), females (8 mg/day) and children (5 mg/day). The Zn contents in the roadside, bush and farm-margin pulp correspond to 1% by UL parameters for males, females, pregnant women (40 mg/day) and children (12 mg/day). The Zn concentrations in this pulp correspond to 2.67%, 2.27% and 3.4% by 3 mg/day FAO/WHO limits [31]. Based on FDA parameters, this pulp is not a good source of Zn [31]. The Zn concentrations in this pulp are lower compared with the amount of 0.2–0.5 mg/100 g reported in previous studies on fruits and pulp of *C. adamantium* [18,19,34]. However, the Zn amounts are between those of apple, grapes and tomato (0.04–0.17 mg/100 g) [38]. The health benefit of the consumption of Zn food is associated with preventing or reducing oxidative stress, infections (malaria, pneumonia and diarrhea), cell ageing, atherosclerosis, neuropsychological diseases, autoimmune and degenerative diseases, Alzheimer’s disease, inflammation cytokine storms, cancers, diabetes mellitus, obesity, depression, gastrointestinal and reproductive organ dysfunction, retina disease, and improving fetal and childhood skeletal growth and development [68,69,70].

Cobalt (Co) concentrations in roadside (0.07 ± 0.01 mg/100 g), bush (0.02 ± 0.00 mg/100 g) and farm-margin (0.08 ± 0.02 mg/100 g) pulp correspond to 5000%, 1428.57% and 5714.29% by 0.0014 mg/day WHO limits [33]. The RDA and UL parameters for Co have no established value for adults and children. Conforming to FDA parameters, this pulp is an excellent source of Co [38]. The Co concentrations in this pulp are lower than 8 mg/100 g reported in previous studies on *C. adamantium* pulp [34]. The Co concentrations are between strawberries, apple, grapes, mango, tomato and orange (0.03–0.08 mg/100 g) [44]. The risk of consuming food with a high amount of Co is correlated with inflammation and hypersensitivity reactions [71], neurological, cardiovascular and endocrine deficiency [72].

Nickel (Ni) concentrations in roadside (0.06 ± 0.01 mg/100 g), bush (0.04 ± 0.01 mg/100 g) and (0.1 ± 0.02 mg/100 g) pulp correspond to 6%, 4% and 10% for males, females, pregnant women, and 20%, 13.33% and 33.33% for children, respectively, by 1 mg/day, 1 mg/day and 0.3 mg/day UL limits. The Ni concentrations of the roadside, bush and farm-margin correspond to 30%, 20% and 50% by 0.2 mg/day FAO/WHO limits [32], respectively. The RDA parameters for Ni has no established value for adults and children. According to FDA parameters, this pulp is an excellent source of Ni [38]. The Ni concentrations in this pulp are lower than 4.2 mg/100 g reported in previous studies on fruits of *C. adamantium* [18]. The Ni concentrations are between those of paw-paw, mango, watermelon and banana fruits (0.023–0.089 mg/100 g) [73]. Some articles reported that the health benefit of Ni food consumption is correlated with gut microbiota balance and welfare [74]. However, other studies correlated Ni with hazardous conditions for human health such as cardiovascular, kidney and lung dysfunctions and oxidative stress [75].

Manganese (Mn) concentrations in roadside (0.05 ± 0.01 mg/100 g), bush (0.03 ± 0.01 mg/100 g) and farm-margin (0.07 ± 0.01 mg/100 g) pulp correspond to values ≤4% for males (2.3 mg/day), females (1.8 mg/day), pregnant women (2.6 mg/day) and children (1.5 mg/day) by RDA parameters. The Mn contents correspond to proportions <2.5% for males/females and pregnant women (11 mg/day), and children (3 mg/day) by UL limits. The Mn concentrations in pulps of roadside, bush and farm-margin correspond to 1.33%, 1.00% and 2.33%, respectively, by 3 mg/day FAO/WHO limits [31]. By FDA parameters, this pulp is not a good source of Mn [38]. The Mn concentrations in this pulp are lower than the amounts of 0.09–0.21 mg/100 g reported in previous studies on fruits and pulp of *C. adamantium* [18,19,34]. However, the Mn contents are near those of paw-paw and wheat (0.08–1.0 mg/100 g) [76]. The health benefit of the consumption of Mn food is associated with gut microbiota balance, regulating oxygen species and anemia conditions between mother and fetus and neurodevelopment [77,78,79].

Chromium (Cr) concentrations in roadside pulp was 0.03 ± 0.01 mg/100 g, which corresponds to 116.67%, 83.33%, 100% and 50% for males (0.035 mg/day), females (0.025 mg/day), pregnant women (0.03 mg/day) and children (0.015 mg/day) by AI parameters, respectively. The Cr content of 0.01 ± 0.00 mg/100 g in bush pulp corresponds to 350%, 250%, 300%, and 150% for males, females, pregnant women and children, respectively, by AI limits. The Cr content in farm-margin pulp was 0.05 ± 0.01 mg/100 g, which corresponds to 70%, 50%, 60%, and 30% for males, females, pregnant women and children, respectively, according to the AI standard. The Cr concentrations in the roadside, bush and farm-margin pulp correspond to 6.67%, 20% and 4%, respectively, by 0.002 mg/day FAO/WHO limits [32]. The RDA and UL parameters for Cr have no established values for adults and children. According to FDA parameters, this pulp is a good source of Cr [38]. However, the Cr concentrations in this pulp are lower than the amount of 0.1–1.14 mg/100 g reported in previous studies on *C. adamantium* pulp [19,34]. The Cr contents are near edible fruits such as strawberry and melon (0.3 mg/100 g) [80].

### 3.2. Health Risk Assessment

The carcinogenic risk (CR) was calculated for three chemical elements Pb, As and Cr in pulp obtained from fruits collected in roadside, bush and farm-margin areas (Table 4). The values of As and Cr were farm-margin > roadside > bush, while the Pb values differed (farm-margin > bush > roadside). The total cancer risk (R) values for farm-margin, roadside and bush were 1.5 × 10^−3^, 1.1 × 10^−3^ and 6.2 × 10^−4^, respectively, which were higher compared with the acceptable parameters (10^−6^ to 10^−4^), showing the importance of these values in terms of their carcinogenic risk for *C. adamantium* pulp consumers of 10 g/kg/day [26]. The total cancer risk is presented in decreased order As > Pb > Cr, demonstrating that As is the main pollutant chemical element that can be correlated with several cancer incidences among all heavy metals found in this pulp. Furthermore, the total cancer risk incidence can be higher for those who consume the recommended intake of 400 g/day [81] of pulp from farm-margin, roadside and bush (6.1 × 10^−2^, 4.2 × 10^−2^ and 2.5 × 10^−2^, respectively), in the region crossed by a road of high large vehicle traffic and intensive modern agriculture. However, the total cancer risks for the consumption of 1 g/kg/day of pulp from the roadside, bush and farm-margin were estimated to 1.1 × 10^−4^, 6.3 × 10^−5^ and 1.5 × 10^−4^, respectively, which are near of within acceptable parameters [26].

The non-carcinogenic risks for chemical elements are summarized in Table 4. The CDI values of the chemical elements in fruit pulp were presented in decreased order for three collection sites: K > Pb > P > As > Fe > Se > Mo > Zn > Co > Ni > Mn > Cr for roadside, K > Pb > P > As > Se > Fe > Mo > Zn > Co > Ni > Mn > Cr for bush, and K > Pb > P > As > Se = Fe > Mo > Zn > Ni > Co > Mn > Cr for farm-margin. The ordered concentrations of chemical elements are different for Fe and Se from roadside, while these are Se, Fe, Ni and Co for the farm-margin compared with bush areas. The major chemical elements in the pulp in decreased order are farm-margin > roadside > bush, which signifies that the farm and road have spread these chemical elements to pollute fruits. In contrast, Pb and Se are ordered from farm-margin > bush > roadside, which explains that the highest amount of these chemical elements have spread from the farm.

The hazard quotient (HQ) values of the chemical elements in roadside pulp estimated in decreased order are As > Pb > Mo > Cr > Ni > Co > Fe > Mn > Zn; in bush pulp: As > Pb> Mo > Cr > Ni > Co > Fe > Zn > Mn; and in farm-margin pulp: As > Pb > Mo > Cr > Ni > Co > Mn > Fe > Zn. The contents of Mn, Fe and Zn are irregularly distributed in farm-margin, roadside and bush areas. The majority of chemical elements were ordered as farm-margin > roadside > bush, which explains that the farm and road are sources of higher amounts of these chemical elements. In contrast, Pb is ordered as farm-margin > bush > roadside, meaning that this chemical element has spread in a higher amount from the farm. The majority of chemical elements presented HQ < 1, while the highest values of As in the farm-margin, roadside and bush were 3.33, 2.30 and 1.34, respectively. Therefore, with a consumption of 10 g/kg/day of pulp, As could be the main cause of several cancer types and other chronic diseases.

## 4. Conclusions

According to RDA and UL limits, the pulp of *C. adamantium* collected in areas located between the road subject to high large vehicle traffic and intensive modern agriculture farming presented the lowest concentration of K, P, Se, Fe, Mo, Zn, Ni, and Mn. However, based on FAO/WHO parameters, the highest concentrations are Pb, As, Se, Mo, Co and Ni, and the lowest are K, P, Fe, Zn and Mn. The Cr concentration is above FAO/WHO and AI limits. Values of Pb, As, Co and Cr are not established by RDA and UL standards, including K, which are not established for the last parameter. This pulp is an excellent source of Pb, As, Se, Mo, Co, Ni and Cr, while it is not a good source of K, P, Fe, Zn and Mn, based on FDA parameters. It is notable that plants that grow and develop between intensive anthropogenic and severe activities are contaminated by heavy metals such as Pb, As, Mo, Co, Ni, Mn and Cr. Additionally, the concentrations of these heavy metals increase, while K, P, Fe and Zn decrease, except Se. Therefore, the consumption of plants collected in these environments can be a hazard to human health. Therefore, toxicological studies may be necessary to guarantee the safe consumption of edible plants collected in areas under intensive severe anthropogenic activities.

Overall, the estimated carcinogenic risk and total cancer risk in this pulp are represented by As, Pb and Cr, which are in higher concentrations in pulp collected in farm-margin, followed by the roadside and bush. The primary crucial heavy metal is As, presenting HQ > 1 (3.33, 2.30 and 1.34 in pulp collected in farm-margin, roadside and bush, respectively). However, quantities ≤ 1 g daily intake of pulp obtained in these areas can decrease the total cancer risk and are within accepted parameters and HQ < 1 for all chemical elements assessed in this pulp. This demonstrated that modern intensive agriculture farms and areas crossed by roads of large vehicle traffic are sources of pollutants that contaminate fruits and vegetables that grow in surrounding areas.

## Figures and Tables

**Figure 1 ijerph-18-05503-f001:**
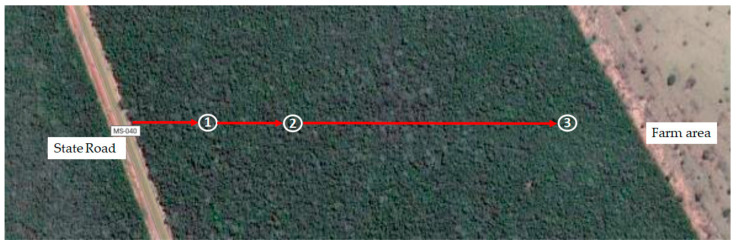
Collection spots of *Campomanesia adamantium* fruits located between the state road MS-040 with high large vehicle traffic and intensive modern agriculture in Campo Grande—Mato Grosso do Sul State, Brazil. 1. roadside = 500 m; 2. bush = 1000 m; and 3. farm-margin = 3000 m.

**Figure 2 ijerph-18-05503-f002:**
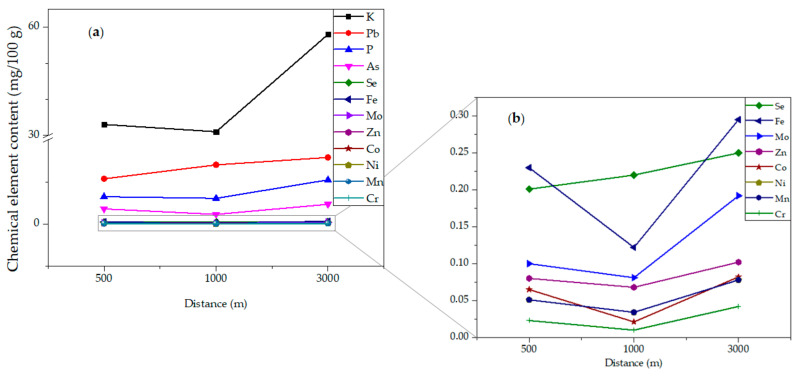
Behavior of the chemical elements’ quantities distribution in *Campomanesia adamantium* pulp collected in three different sites: roadside (500 m); bush (1000 m); and farm-margin (3000 m), quantified by ICP OES (mg/100 g): (**a**) chemical element content >1 mg/100 g; (**b**). chemical element content ≤0.4 mg/100 g.

**Table 1 ijerph-18-05503-t001:** Microwave digestion parameters.

	Steps			
	1	2	3	4
Power (W)	1305	1305	0	0
Temperature (°C)	170	200	50	50
Ramp time (min)	1	1	1	1
Hold time (min)	10	15	10	1
Pressure (Bar)	35	35	0	0

**Table 2 ijerph-18-05503-t002:** Instrumental analytical conditions for the ICP OES of element analysis.

Parameters	Setting
RF power (W)	1250
Sample flow rate (L mn^−1^)	0.45
Plasma gas flow rate (L mn^−1^)	12
Integration time (s)	5
Stabilization time (s)	20
Pressure of nebulization (p si)	20
Plasm view	Axial
Gas view	Air
Analytical wavelength (nm)	Fe (259.940), Ni (231.604), Co (228.616), Cr (267.716), As (193.759), Pb (214.441), Mo (202.030), Mn (257.610), P (177.595), K (766.490), Zn (213.856), Se (196.090).

**Table 3 ijerph-18-05503-t003:** *Campomanesia adamantium* pulp collected in three different sites from the road: roadside (500 m); bush (1000 m); and farm-margin (3000 m), quantified by ICP OES (mg/100 g ± SD) compared with nutritional recommendations for adult, pregnancy and children by RDA/AI, UL and FAO/WHO.

Elements	Roadside (mg/100 g)	Bush (mg/100 g)	Farm-Margin (mg/100 g)	Male 31–50 y RDA/AI * (mg/day)	Female 31–50 y RDA/AI * (mg/day)	Male/ Female 31–50 y UL (mg/day)	Pregnancy 31–50 y		Children 4–8 y		FAO/WHO (mg/day)
							RDA/AI * (mg/day)	UL (mg/day)	RDA/AI * (mg/day)	UL (mg/day)	
K	33.02 ± 0.01	31.02 ± 0.01	58.01 ± 1.34	4700	4700	ND	4700	ND	4700	ND	3510 [29]
Pb	5.36 ± 0.02	7.02 ± 0.01	6.85 ± 1.05	ND	ND	ND	ND	ND	ND	ND	0.02 [30]
P	3.24 ± 0.02	3.04 ± 0.02	5.24 ± 0.80	700	700	4000	700	3500	500	3000	700 [29,31]
As	1.96 ± 0.04	1.14 ± 0.03	2.84 ± 0.52	ND	ND	ND	ND	ND	ND	ND	0.01 [30]
Se	0.20 ± 0.01	0.22 ± 0.02	0.40 ± 0.10	55	55	400	400	60	30	150	0.06 [31]
Fe	0.23 ± 0.02	0.12 ± 0.01	0.40 ± 0.10	8	18	45	27	45	10	40	2.00 [32]
Mo	0.10 ± 0.02	0.09 ± 0.02	0.19 ± 0.01	150	150	1100	50	2000	22	600	0.045 [31]
Zn	0.08 ± 0.01	0.07 ± 0.01	0.13 ± 0.02	11	8	40	11	40	5	12	3.00 [31]
Co	0.07 ± 0.01	0.02 ± 0.00	0.08 ± 0.02	ND	ND	ND	ND	ND	ND	ND	0.0014 [33]
Ni	0.06 ± 0.01	0.04 ± 0.01	0.10 ± 0.02	ND	ND	1	ND	1	ND	0.3	0.20 [32]
Mn	0.05 ± 0.01	0.03 ± 0.01	0.07 ± 0.01	2.30	1.80	11	2.60	11	1.50	3	3.00 [31]
Cr	0.03 ± 0.01	0.01 ± 0.00	0.05 ± 0.01	0.035 *	0.025 *	ND	0.030 *	ND	0.015 *	ND	0.002 [32]

Note. ND—not determined; * The value for AI is used when there are no calculated values for the RDA.

**Table 4 ijerph-18-05503-t004:** Carcinogenic risk (CR), chronic daily intake dose (CDI, mg/kg bw/day) and hazard quotient (HQ) of chemical elements based on 10 g of *Campomanesia adamantium* pulp collected at three different sites from the road: roadside (500 m), bush (1000 m) and farm-margin (3000 m).

Samples		K	Pb	P	As	Se	Fe	Mo	Zn	Co	Ni	Mn	Cr
Roadside	CR	-	0.000016	-	0.001036	-	-	-	-	-	-	-	0.0000053
	CDI	0.011631	0.001888	0.001141	0.000690	0.000070	0.000081	0.000035	0.000028	0.000025	0.000021	0.000018	0.000011
	HQ	-	0.472016	-	2.301370	-	0.000116	0.007045	0.000094	0.000822	0.001057	0.000126	0.003523
Bush	CR	-	0.000021	-	0.000602	-	-	-	-	-	-	-	0.0000018
	CDI	0.010927	0.002473	0.001071	0.000402	0.000077	0.000042	0.000032	0.000025	0.000007	0.000014	0.000011	0.000004
	HQ	-	0.618200	-	1.338552	-	0.000060	0.006341	0.000082	0.000235	0.000705	0.000076	0.001174
Farm-margin	CR	-	0.000024	-	0.001501	-	-	-	-	-	-	-	0.0000088
	CDI	0.020434	0.002779	0.001846	0.001000	0.000141	0.000141	0.000067	0.000046	0.000028	0.000035	0.000025	0.000018
	HQ	-	0.694814	-	3.334638	-	0.000201	0.013386	0.000153	0.000939	0.001761	0.001233	0.005871

## Data Availability

Data will be available upon reasonable request to corresponding author.

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
