# Peer review of "High Concentration of Heavy Metal and Metalloid Levels in Edible Campomanesia adamantium Pulp from Anthropic Areas"

_ijerph, 2021, doi:10.3390/ijerph18115503_

Round 1

Reviewer 1 Report

The manuscript describes study, quite interesting, related to evaluation of the content of heavy metal, non-metal and metalloid level of  Campomanesia adamantium pulp, after contamination exposition. The manuscript can be accepted after considering the comments below:

  • Page 1, Line 35: The statement “Native fruits contribute to food security“ is not clear.
  • Page 2, Line 60: The objective and novelty of the work should be emphasized.
  • Page 3, Table 1: No differences were found between the steps 3 and 4 of the digestion procedure. Is it correct?.
  • Page 3, Line 92: If I have understood properly, samples were diluted up 100 ml after digestion (line 84). However, in this paragraph, it is indicated that samples were transferred to 50 ml Falcon and then diluted with 30 ml of water. This should be clarified because the sample volume is higher than 30 ml.
  • Page 3, Line 90: Was an internal standard used? Or a quality control sample? Just to be sure that no signal-drift over time was observed.
  • Page 3, Lines 93-95: The phrase should be rewritten.
  • Page 4, Line 104: I would indicate procedural LOD and LOQ instead of LOD/LOQ of de ICP-OES determination.
  • Page 6, Table 3: Data should be reviewed, significal figures are not properly used.
  • Page 7, Line 176: When you compare with previous work, you should take not account if the exposition conditions to contaminants are the same or not. Please, clarify.
  • Page 7, Line 187: Is not clear. The information appears to be contrary to the information of the line 182.
  • General comments: More emphasis should be placed on the novelty of this work. Moreover, do you have more information about other metals and metalloids? And about more sampling points? The representativity of only three sampling points is not too high.

Author Response

Dear Reviewer

We thank for your suggestions to improve our manuscript.

All your suggestions and questions on manuscript are in attachment.

All authors.

Reviewer 2 Report

Thanks for your submission. 

The first question, you need to clearly justify why do you select Campomanesia adamantium. 

The introduction is just focused on the selected area, please try to emphasis the overall importance of the combination between food security and heavy metal pollutions in general. 

Please kindly refer 

Sustainable Water Resource Management for a Safe Food System in Georgia: A Study of Water Quality Governance, focusing the Mashavera River Basin (uni-kassel.de)  

As you study human health risk effects, the examination of risk perception is widely important. Please refer to Sustainability | Free Full-Text | Farmers’ Perception of Water Quality and Risks in the Mashavera River Basin, Georgia: Analyzing the Vulnerability of the Social-Ecological System through Community Perceptions (mdpi.com)

Line 209-211, why K concentration in pulp is low in your field area? need to explain 

With Table 3, I would propose to have an interactive graphical analysis for three field areas. 

Figure 2 is not clearly presented. 

It would be important to find the heavy metal concentration in the soil samples too. Do you concentrate only on the dust particle? 

Author Response

(The authors gave the same response as above.)

Round 2

Reviewer 1 Report

The manuscript can be accepted after considering the comments below:

  • Page 2, Line 97: Please unify the criteria.Some elemets are named with the full name and others with the abbreviation.
  • Page 2, Line 101: “The concentration of these minerals compared with rec” should be replaced by “The concentration of these minerals was compared with rec”.
  • Page 3, Line 156-157: “The samples were digestion steps according to the schedule shown in Table 1” should be replaced by “The samples were digested according to the schedule shown in Table 1”
  • Page 4, Line 196: “were obtained multiple-element stock solutions containing”, this sentence should be corrected.
  • Page 4, Line 204: “sign” should be replaced by “signal”.
  • Page 15, Line 887: “presents” should be replaced by “presented”.

Author Response

Dear Reviewer

We are glade to return to you our manuscript. We would like thank you for your suggestions and observations you done to improve our work. English writing was reviewed throughout the text.

Best Regards
The authors

Reviewer 2 Report

Thanks for your revised version. I would like to accept the manuscript. 

Author Response

Dear Reviewer

We glade to return to you our manuscript. We want thank you for your brilliant suggestions and recommendations that were crucial to improve our work. Writing English language was reviewed throughout the manuscript.

Best Regards

The authors
